# Diagnostic potential of tumor DNA from ovarian cyst fluid

Yuxuan Wang[1,2†], Karin Sundfeldt[3†], Constantina Mateoiu[4], Ie-Ming Shih[5,6], Robert J Kurman[5,6], Joy Schaefer[1,2], Natalie Silliman[1,2], Isaac Kinde[1,2], Simeon Springer[1,2], Michael Foote[1,2,7], Björg Kristjansdottir[3], Nathan James[1,2], Kenneth W Kinzler[1,2], Nickolas Papadopoulos[1,2], Luis A Diaz[1,2,7], Bert Vogelstein[1,2]*

[1]Ludwig Center, Howard Hughes Medical Institute, Johns Hopkins University, Baltimore, United States; [2]Sidney Kimmel Comprehensive Cancer Center, Howard Hughes Medical Institute, Johns Hopkins University, Baltimore, United States; [3]Department of Obstetrics and Gynecology, Institute of Clinical Sciences, Sahlgrenska Cancer Center, University of Gothenburg, Gothenburg, Sweden; [4]Department of Pathology, Institute of Biomedicine, University of Gothenburg, Gothenburg, Sweden; [5]Department of Pathology, The Johns Hopkins Medical Institutes, Baltimore, United States; [6]Department of Gynecology and Obstetrics, The Johns Hopkins Medical Institutions, Baltimore, United States; [7]Swim Across America Laboratory, Baltimore, United States

**Abstract** We determined whether the mutations found in ovarian cancers could be identified in the patients' ovarian cyst fluids. Tumor-specific mutations were detectable in the cyst fluids of 19 of 23 (83%) borderline tumors, 10 of 13 (77%) type I cancers, and 18 of 18 (100%) type II cancers. In contrast, no mutations were found in the cyst fluids of 18 patients with benign tumors or non-neoplastic cysts. Though large, prospective studies are needed to demonstrate the safety and clinical utility of this approach, our results suggest that the genetic evaluation of cyst fluids might be able to inform the management of the large number of women with these lesions.

*For correspondence: bertvog@gmail.com

†These authors contributed equally to this work

## Introduction

Ovarian cancer is the most lethal gynecologic malignancy, with 21,290 estimated new cases and 14,180 estimated deaths in the United States in 2015. Approximately 1.3% of women will be diagnosed with ovarian cancer during their lifetime (*Howlader et al., 2014*). These cancers commonly present as an adnexal mass with cystic components, but are not associated with specific symptoms. As a result, two-thirds of ovarian cancers are diagnosed at late stage (Stage III or IV), when the 5-year survival is less than 30% (*Howlader et al., 2014*)

Complicating the diagnosis of ovarian cancer is the fact that ovarian cysts are common in women of all ages, with a prevalence of 35% and 17% in pre- and post-menopausal women, respectively (*Pavlik et al., 2013*). These cysts are frequently benign and found incidentally on routine imaging (*Pavlik et al., 2013*). Though malignancy is an unusual cause of the cysts, 30% of the cysts exhibit radiographic features suspicious for malignancy, such as solid areas or mass (*Pavlik et al., 2013*). In addition to the anxiety that such findings provoke, many women undergo unnecessary surgery for cysts that are not malignant and may not be responsible for the symptoms they have. For example, only 5% of 570 women in a large ovarian cancer screening randomized trial who underwent surgical evaluation actually had a malignancy (*Buys et al., 2005*). Similarly in another study of symptomatic

**eLife digest** More than a third of women develop ovarian cysts during their lifetimes. The vast majority of these cysts are harmless, but a small number are caused by ovarian cancers. These cancers often produce no symptoms until the disease has spread throughout the abdomen or to other organs, so many women go undiagnosed until their chances of being successfully treated are low. Currently, there is no reliable way to determine whether an ovarian cyst is cancerous without performing surgery. As a result, many women undergo unnecessary, invasive surgeries for harmless ovarian cysts.

Tumors shed cells and cell fragments into any fluid that surrounds them. Fluids from cysts in the pancreas, kidney, and thyroid are routinely examined to identify whether they contain cancerous cells. Now, Wang, Sundfeldt et al. show that ovarian cancers also shed DNA into the surrounding cyst fluid. Furthermore, mutations found in this DNA can provide valuable information about whether the cysts are cancerous.

The study was performed by extracting DNA from the fluid in ovarian cysts that had been surgically removed from 77 women. Of these cysts, 10 were harmless cysts, 12 were benign tumors, 31 were invasive cancers, and 24 were so-called borderline tumors, which fall somewhere between the benign tumors and invasive cancers. Only cysts associated with the borderline tumors and invasive cancers need to be surgically removed. Here, Wang, Sundfeldt et al. report that DNA mutations that are characteristic of ovarian cancers were found in 87% of the cysts associated with borderline tumors and invasive cancers. In contrast, these mutations were not found in any of the cysts that do not require surgery.

Fluid can be extracted from an ovarian cyst with a needle during an outpatient visit. Therefore, the results presented by Wang, Sundfeldt et al. suggest a relatively straightforward way of testing the DNA from ovarian cysts before deciding whether surgery is really necessary. First, however, larger studies that follow women with cysts over time will be necessary to confirm that this type of testing is effective and safe.

women, only 4% of 197 women who had concerning features on transvaginal ultrasound were ultimately diagnosed with ovarian cancer (*Gilbert et al., 2012*). Compounding this issue is the fact that surgery for ovarian cysts requires general anesthesia and is associated with significant morbidity, causing complications in 15% of women (*Buys et al., 2011*). These complications include damage to nerves and ureters, bleeding, infection, and perforation of adjacent viscera. Furthermore, the procedure often results in hormonal and fertility loss (in the case of bilateral oophorectomy) (*Buys et al., 2011*). Even minimal procedures such as ovarian cystectomy can affect fertility in premenopausal women by decreasing follicular response and oocyte number (*Loh et al., 1999*; *Demirol et al., 2006*). If a preoperative test could help determine whether the cystic lesion was benign or malignant, unnecessary surgery and its associated complications could be avoided in many patients. This would be particularly helpful for women of reproductive age who wish to preserve their fertility, as well as women whose medical comorbidities or functional status makes anesthesia and surgery hazardous.

Ovarian cysts and tumors are classified as non-neoplastic, benign, borderline, or malignant based on microscopic examination after surgical removal (*Figure 1*). Non-neoplastic cysts are by far the most common class of ovarian cysts. They are frequently found in pre-menopausal women, arising when an egg is not released properly from either the follicle or corpus luteum and usually resolve spontaneously within several months (*Christensen et al., 2002*). Benign cystic tumors, such as cystadenomas and cystadenofibromas, rarely progress to malignancy (*Cheng et al., 2004*; *Levine et al., 2010*). No genetic alterations have yet been identified in either non-neoplastic cysts or in benign cystic tumors (*Cheng et al., 2004*). Neither of these cyst types requires surgery unless they are symptomatic or large (*Levine et al., 2010*). These cysts can be easily sampled with ultrasound-guided fine-needle aspiration within minutes in an outpatient setting without the need for anesthesia (*Duke et al., 2006*).

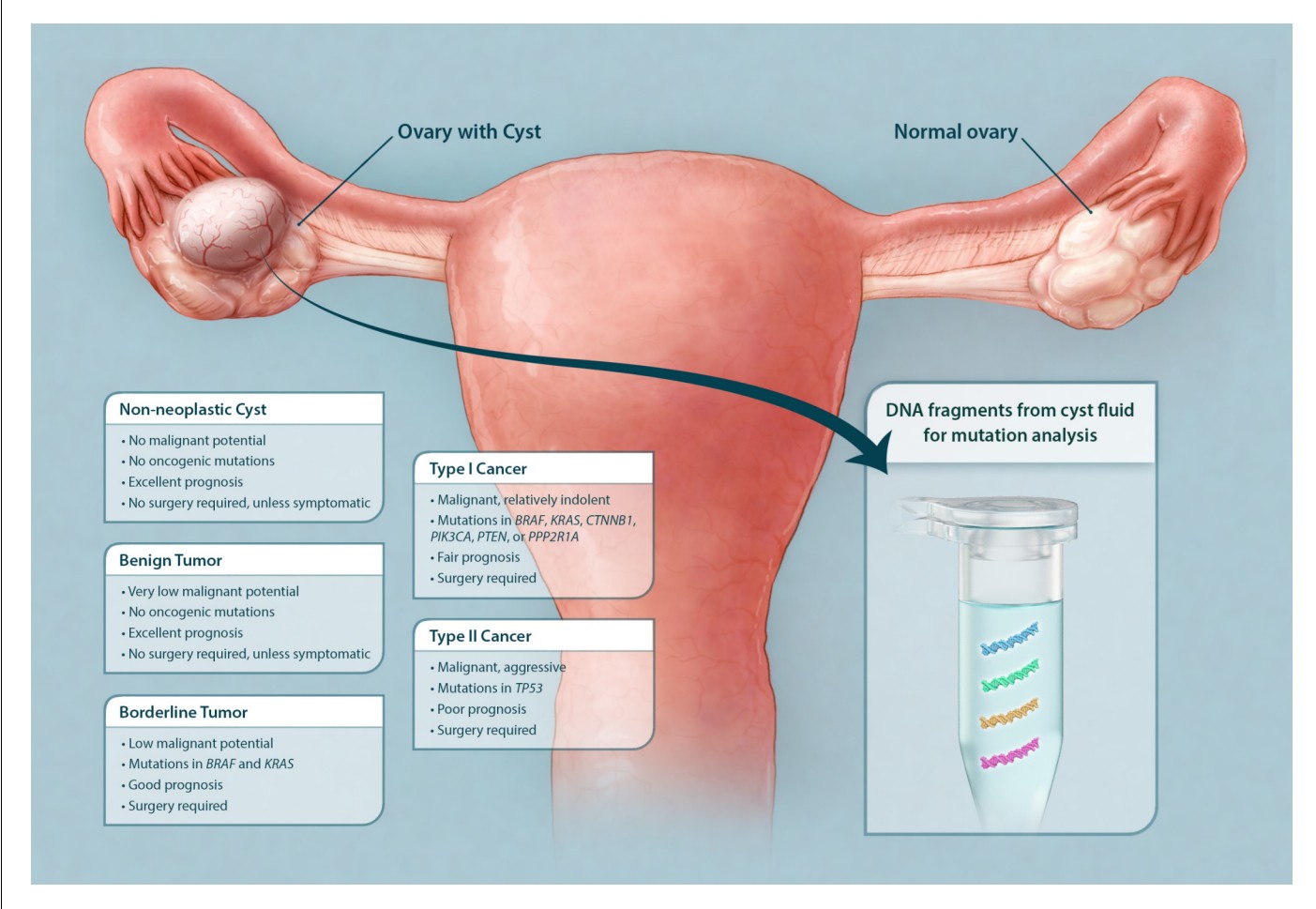

**Figure 1.** Schematic showing classes of ovarian cysts and the diagnostic potential of the cyst fluid. Ovarian cysts and tumors are currently classified according to microscopic evaluation after surgical removal. The majority of ovarian cysts are non-neoplastic (often 'functional' in premenopausal women). Ovarian tumors with combined cystic and solid components are either benign tumors, borderline tumors, or malignant cancers (type I or II). Only cysts associated with borderline tumors and cancers require surgical excision. We show here that the DNA purified from cyst fluid can be analyzed for somatic mutations commonly found in their associated tumors. The type of mutation detected not only could indicate the type of tumor present but also could potentially inform management.

At the other end of the spectrum are epithelial ovarian cancers, which are potentially lethal and unequivocally require surgery. A dualistic model has been proposed to classify these neoplasms (*Kurman and Shih, 2010*). Type I tumors are composed of low-grade serous, low-grade endometrioid, clear cell, and mucinous carcinomas. They are generally indolent, frequently diagnosed at early stage (Stage I or II), and develop from well-established precursor lesions ('borderline' or 'atypical proliferative' tumors, as described below) (*Kurman and Shih, 2011*). Type I cancers commonly exhibit mutations in *KRAS, BRAF, CTNNB1, PIK3CA, PTEN, ARID1A*, or *PPP2R1A* (*Kurman and Shih, 2010*). In contrast, type II tumors are generally high-grade serous carcinomas. They are highly aggressive, most often diagnosed in late stage (Stage III or IV), and have suggested origins from the distal fallopian tube (*Lee et al., 2007*). Type II cancers are the most clinically important group of ovarian cancers, comprising 75% of all ovarian carcinomas and responsible for 90% of ovarian cancer deaths (*Kurman and Shih, 2011*). They almost always harbor *TP53* mutations (*Cancer Genome Atlas Research Network. 2011*). Also unlike type I cancers, which are relatively chemo-resistant and more often treated only with surgical excision, type II cancers respond to conventional chemotherapy, particularly after maximal debulking to reduce tumor burden (*Bristow et al., 2002*; *Schmeler et al., 2008*).

**Table 1.** Detection of tumor-specific mutations in cyst fluid. The fraction of samples detected and the median fraction of mutant alleles are indicated, grouped by cyst type, cancer stage, and the need for surgery.

| | Fraction of samples detected (95% confidence interval) | Median fraction of mutant alleles (IQR) | Total # of samples |
|---|---|---|---|
| *Type* | | | |
| Non-neoplastic | 0% (0–46%) | 0% (0–0%) | 6 |
| Benign tumor | 0% (0–26%) | 0% (0–0%) | 12 |
| Borderline tumor | 83% (61–95%) | 2.4% (1.5–10.8%) | 23 |
| Type I cancer | 77% (46–95%) | 7.8% (3.3–28.7%) | 13 |
| Type II cancer | 100% (81–100%) | 60.3% (31.3–70.8%) | 18 |
| *Cancer stage* | | | |
| Early (I and II) | 82% (48–97%) | 7.4% (3.0–30.9%) | 11 |
| Late (III and IV) | 95% (75–100%) | 51.2% (30.2–69.5%) | 20 |
| *Cysts requiring surgery* | | | |
| No | 0% (0–26%) | 0% (0–0%) | 18 |
| Yes | 87% (75–95%) | 12.6% (2.7–40.2%) | 54 |

'Borderline' or 'atypical proliferative' tumors lie in the middle of this spectrum, between the malignant cancers and the generally harmless non-neoplastic or benign lesions. They are distinguished from carcinomas by the absence of stromal invasion and are precursors of type I cancers. In light of their potential for malignancy, the standard of care for borderline tumors is surgical excision. Following surgery, the prognosis is excellent compared to ovarian cancers, with 10-year survival rates over 94% (*Sherman et al., 2004*). A minor but significant portion of borderline tumors recur after surgery, however, and a subset of the recurrences are found to have advanced to type I cancers (*Shih et al., 2011*). This progression is consistent with molecular findings: serous borderline tumors typically exhibit mutations in *BRAF* or *KRAS*, like their malignant counterparts (low-grade serous carcinoma) (*Mayr et al., 2006*; *Jones et al., 2012*). The presence of a *BRAF* mutation in a borderline tumor is associated with better prognosis and a low probability of progression to carcinoma (*Grisham et al., 2013*). In contrast, *KRAS* mutations are associated with the progression to type I cancers (*Tsang et al., 2013*).

The examination of fluids from pancreatic, renal, and thyroid cysts is routinely used in clinical management (*Frossard et al., 2003*; *Lin et al., 2005*; *Volpe et al., 2007*). The fluids have historically been studied by cytology to identify malignant cysts. Ovarian cysts share many features with these other types of cysts, in that they are common, often diagnosed incidentally, and are nearly always benign. However, aspiration of ovarian cyst fluid for cytology is not standard-of-care. From a historical perspective, the difference in diagnostic management partly lies in the fact that cytology has not proven to be very informative for ovarian cysts, particularly for distinguishing benign vs. borderline tumors (*Moran et al., 1993*; *Martínez-Onsurbe et al., 2001*). There have also been concerns raised

**Table 2.** Multivariate analysis for markers associated with need for surgery. The presence of a mutation, cyst DNA amount, and common serum biomarkers for ovarian cancer were analyzed for association with cysts that require surgical removal (Firth's penalized likelihood logistic regression).

| Criteria | p value |
|---|---|
| Mutation present | <0.001 |
| Serum CA-125 elevated | 0.01 |
| HE4 elevated | 0.92 |
| Cyst DNA amount | 0.69 |

about the safety of ovarian cyst aspiration (see Discussion), and these concerns have not often been raised for other types of cysts.

More recently, genetic analysis of specific types of cyst fluids has been considered as an aid to cytology, given that conventional cytology often has limited sensitivity and specificity (*Wu et al., 2011b*). Based on the emerging success of the molecular genetic evaluation of other types of cysts, we reasoned that a similar approach could be applied to ovarian cysts. Evaluation of DNA from cells and cell fragments shed into the cyst fluid would presumably allow the identification of tumor-specific mutations. Unlike other, conventional markers of neoplasia such as CA-125, cancer gene mutations are exquisitely specific indicators of a neoplastic lesion (*Vogelstein et al., 2013*). Moreover, the type of mutation can in some cases indicate the type of neoplastic lesion present (*Wu et al., 2011a*). Yamada et al. have demonstrated that mutations can be detected in the cystic fluid of ovarian tumors by querying exons 4 to 9 of *TP53*, achieving sensitivities of 12.5% and 10%, for borderline and malignant tumors, respectively (*Yamada et al., 2013*). Recently developed, extremely sensitive methods for mutation detection, capable of identifying one mutant template allele among thousands of normal templates in a panel of genes, could potentially increase this sensitivity (*Kinde et al., 2011*; *Murtaza et al., 2013*; *Newman et al., 2014*). In this study, we have applied one of these technologies to assess mutations in ovarian cyst fluids and to inform the development of tests that could eventually be applied to patients.

## Results

### Characteristics of the tumors and cyst fluid samples

DNA was isolated from surgically excised ovarian cysts of 77 women. Ten of them had non-neoplastic cysts, 12 had benign tumors, 24 had borderline tumors, and 31 had cancers (13 Type I and 18 Type II). Age, histopathologic diagnosis, stage, and other clinical information are provided in *Supplementary file 1*. The median amount of DNA recovered from the cysts was 222 ng (interquartile range (IQR) of 53 to 3120 ng) (*Supplementary file 2*). There was no significant difference in the amounts of DNA between borderline tumors and type I or type II cancers. However, the borderline tumors and cancers contained significantly more DNA than the non-neoplastic cysts or benign tumors (4453 ± 6428 ng vs. 62 ± 64 ng; p<0.001, Wilcoxon rank-sum test).

### A multiplex PCR-based assay to identify tumor-specific mutations in cyst fluid samples

We designed a multiplex PCR-based assay that could simultaneously assess the regions of 17 genes frequently mutated in ovarian tumors. The amount of DNA shed from neoplastic cells was expected to be a minor fraction of the total DNA in the cyst fluid, with most DNA emanating from normal cells. We therefore used a sensitive detection method, called Safe-SeqS (Safe-Sequencing System), to identify mutations in cyst fluid samples (*Kinde et al., 2011*). In brief, primers were designed to amplify 133 regions, covering 9054 distinct nucleotide positions within the 17 genes of interest (*Supplementary file 3*). Three multiplex PCR reactions, each containing non-overlapping amplicons, were then performed on each sample. One primer in each pair included a unique identifier (UID) for each template molecule, thereby drastically minimizing the error rates associated with PCR and sequencing, as described previously (*Kinde et al., 2011*). Under the conditions used in the current experiments, mutations present in >0.1% of template molecules could generally be reliably determined (*Kinde et al., 2011*; *2013*; *Bettegowda et al., 2014*). We could not perform sequencing on five cysts (four non-neoplastic cysts and one cyst associated with a borderline tumor) because there was insufficient DNA (<3 ng recovered), and these were scored as uninterpretable. When this assay was applied to cyst fluid samples with sufficient DNA, no mutations were identified in the 18 cysts obtained from patients with simple cysts (n = 6) or benign tumors (n = 12) (*Table 1*). This was in stark contrast to the fluids obtained from the 18 patients with type II cancers, all of which were found to contain a mutation (*Table 1*). Ten (77%) of the 13 cyst fluids from patients with type I cancers and 19 (83%) of the 23 cyst fluids from patients with borderline tumors contained at least one detectable mutation. When categorized by the need for surgery (i.e., presence of a borderline tumor or a type I or type II cancer), the sensitivity of this assay was 87% (47 of 54 cysts; 95% confidence interval of 75% to 95%) and its specificity was 100% (95% confidence interval of 74% to 100%; *Table 1*).

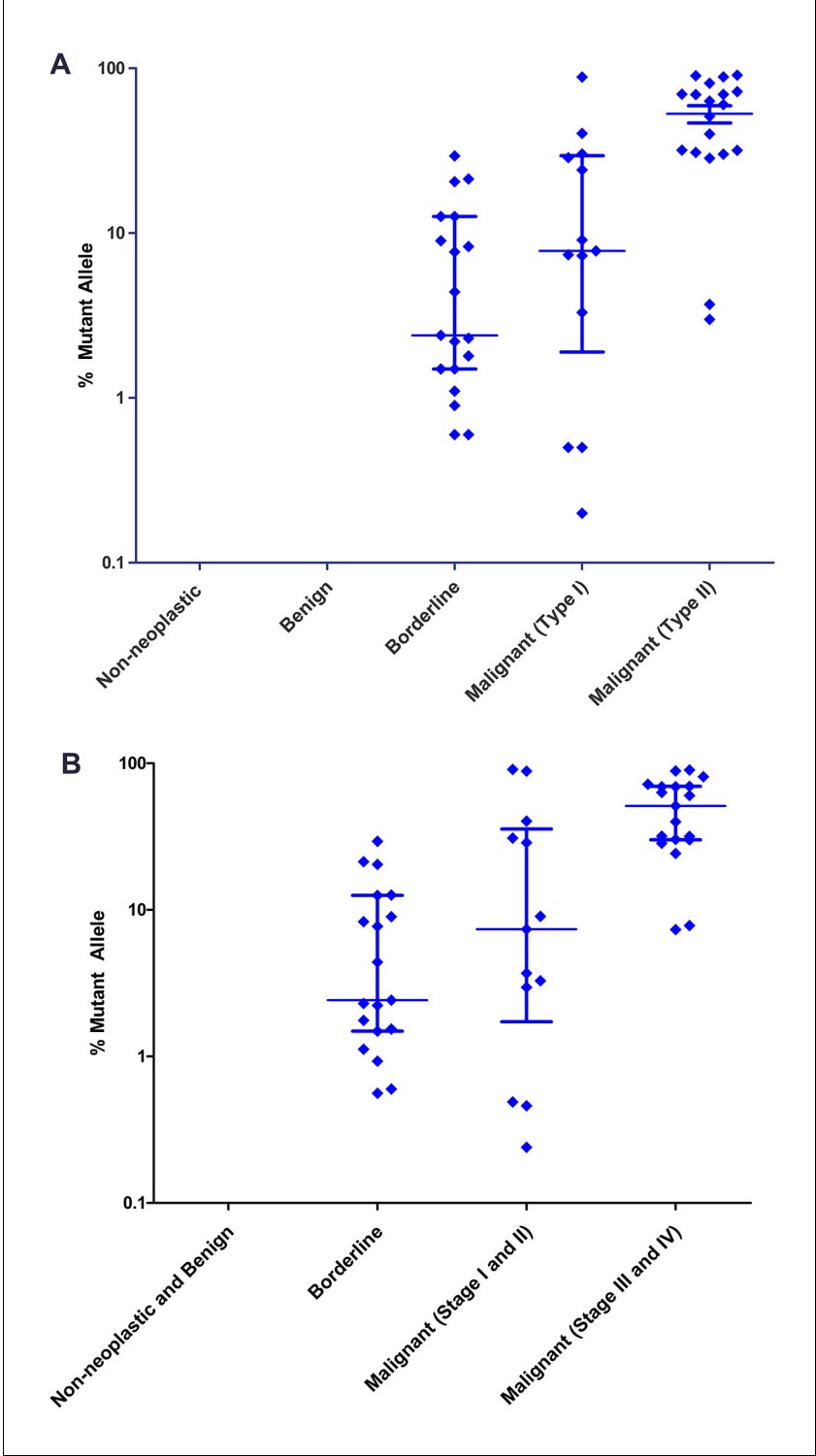

**Figure 2.** Mutant allele fractions. (**A**) Classification by tumor type. No mutations were found in the DNA of non-neoplastic or benign cysts. Of the cysts that required surgery, the median mutant allele fraction was higher in the cyst fluids associated with type II cancer (60.3%) than type I (7.8%) or borderline tumors (2.4%). (**B**) Classification by tumor stage. The DNA from cyst fluids of late-stage cancers had a higher median mutant allele fraction (51.2%) than those of early-stage cancers (7.4%) or borderline tumors (2.4%). Percent mutant allele is depicted on a logarithmic scale. Horizontal bars depict median and IQR.

Ovarian cancers are generally detected only late in the course of disease, perhaps explaining the poor prognosis of patients. Accordingly, only 11 of the 31 cysts associated with cancers in our study had early (Stage I or II) disease (*Supplementary file 1*). As expected, most of these were type I carcinomas (n = 8). Nevertheless, it was encouraging that mutant DNA could be detected in nine (82%) of these 11 patients (*Table 1*). Mutations could be detected in 95% of the 20 patients with Stage III or IV cancers (*Table 1*).

A variety of control experiments were performed to confirm the integrity of these results. One informative positive control was provided by the analysis of DNA from the tumors, using the identical method used to analyze DNA from the cyst fluids. Fifty-three of the 55 borderline and malignant cases had tumors available for this purpose. Every mutation identified in a tumor was found in its cyst fluid, and vice versa. As expected, the mutant allele frequencies in the tumors were often, but not always, higher than in the cyst fluids (*Supplementary file 2*). As another positive control, we used an independent PCR and sequencing reaction to confirm each of the cyst fluid mutations listed in *Supplementary file 2*. This validated not only the presence of a mutation, but also confirmed its fractional representation. The median relative difference between the fractions of mutant alleles in replicate experiments was 7.0% (IQR of 3.5% to 8.9%). Finally, four patients were found to have two independent mutations (*Supplementary file 2*). For example, the cyst fluid of patient OVCYST 081, who had a high-grade endometrioid carcinoma, harbored a missense mutation (R280K) in *TP53* plus an in-frame deletion of *PIK3R1* at codons 458 and 459. The *TP53* mutation was found in 3.0% of alleles while the *PIK3R1* mutation was found in 3.7% of the alleles analyzed. Similar mutant allele frequencies among completely different mutations in the cyst fluid of three other patients provided further indicators of reproducibility. All genetic assays were performed in a blinded manner, with the operator unaware of the diagnoses of the patients from whom the cyst fluids were obtained.

In addition to DNA from normal individuals used as controls, additional negative controls were provided by the simple cysts and benign tumors. Using the identical assay, none of the DNA from their cyst fluids contained detectable mutations. A final control was provided by the borderline and malignant tumors themselves. In general, only one or two of the 9054 base-pairs (bp) queried were mutated in any one *tumor* (*Supplementary file 2*). The other ~9000 bp could then be independently queried in the corresponding *cyst* fluid, and none of these positions were found to be mutated.

## Relationship between the type of tumor present and the type of mutation found in the associated cyst fluid sample

The mutant allele fractions in the cyst fluids tended to be higher in the type II cancers (median of 60.3%) than the type I cancers (median of 7.8%) or borderline tumors (median of 2.4%), though there was considerable overlap (*Table 1*; *Figure 2A*). With respect to stage, the DNA from cyst fluids of late-stage cancers had higher median mutant allele fractions (51.2%) than those of early-stage cancers (7.4%) or borderline tumors (2.4%), but with considerable overlap (*Table 1*; *Figure 2B*).

On the other hand, the type of mutation varied considerably among these cysts (*Figure 3*). In type I tumors, the genes mutated were *BRAF* (n = 1), *KRAS* (n = 5), *NRAS* (n = 1), *PIK3R1* (n = 1), *PPP2R1A* (n = 1), *PTEN* (n = 1), or *TP53* (n = 3). Two distinct mutations were found per sample in three type I cancers. The *BRAF* mutation (V600_S605 > D) was unusual that it resulted from an in-frame deletion/insertion rather than the base substitution (V600E) characteristic of the vast majority of *BRAF* mutations reported in the literature. This mutation has been observed in a papillary thyroid cancer and a cutaneous melanoma (*Cruz et al., 2003*; *Barollo et al., 2014*). The deletion results in loss of a phosphorylation site in the activation loop of BRAF, while the insertion of an aspartic acid has been suggested to increase BRAF kinase activity by mimicking an activating phosphorylation (*Davies et al., 2002*). In contrast, all but one type II cancers (94% of 18) had mutations in *TP53*; the only exception was OVCYST 073, a high-grade endometrioid carcinoma. The borderline tumors were distinguished by yet a different pattern from that of the either type I or type II cancers. Of the 19 mutations in borderline tumors, 12 (63%) were *BRAF* V600E, never observed in type I or type II cancers, and the remainder were at *KRAS* codon 12 or 61 (*Supplementary file 2*).

## Markers associated with the need for surgery

A multivariate analysis was used to identify the most informative molecular features of cyst fluids and to compare them to the commonly used serum biomarkers for ovarian cancer, HE4 (human

| Cyst Classification | Patient ID | BRAF | KRAS | NRAS | PIK3CA | PIK3R1 | PPP2R1A | PTEN | TP53 |
|---|---|---|---|---|---|---|---|---|---|
| Non-neoplastic | *OVCYST 001 | | | | | | | | |
| | *OVCYST 003 | | | | | | | | |
| | OVCYST 004 | | | | | | | | |
| | OVCYST 005 | | | | | | | | |
| | OVCYST 006 | | | | | | | | |
| | OVCYST 007 | | | | | | | | |
| | OVCYST 012 | | | | | | | | |
| | OVCYST 014 | | | | | | | | |
| | *OVCYST 016 | | | | | | | | |
| | *OVCYST 021 | | | | | | | | |
| Benign | OVCYST 002 | | | | | | | | |
| | OVCYST 009 | | | | | | | | |
| | OVCYST 011 | | | | | | | | |
| | OVCYST 008 | | | | | | | | |
| | OVCYST 013 | | | | | | | | |
| | OVCYST 015 | | | | | | | | |
| | OVCYST 017 | | | | | | | | |
| | OVCYST 018 | | | | | | | | |
| | OVCYST 019 | | | | | | | | |
| | OVCYST 020 | | | | | | | | |
| | OVCYST 044 | | | | | | | | |
| | OVCYST 056 | | | | | | | | |
| Borderline | OVCYST 042 | yellow | | | | | | | |
| | OVCYST 043 | | orange | | | | | | |
| | OVCYST 045 | orange | | | | | | | |
| | OVCYST 047 | orange | | | | | | | |
| | *OVCYST 048 | | | | | | | | |
| | OVCYST 049 | red | | | | | | | |
| | OVCYST 050 | yellow | | | | | | | |
| | OVCYST 051 | | | | | | | | |
| | OVCYST 053 | red | | | | | | | |
| | OVCYST 054 | | orange | | | | | | |
| | OVCYST 055 | | red | | | | | | |
| | OVCYST 057 | | orange | | | | | | |
| | OVCYST 058 | | | | | | | | |
| | OVCYST 059 | orange | | | | | | | |
| | OVCYST 060 | red | | | | | | | |
| | OVCYST 061 | | orange | | | | | | |
| | OVCYST 062 | orange | | | | | | | |
| | OVCYST 063 | | orange | | | | | | |
| | OVCYST 064 | yellow | | | | | | | |
| | OVCYST 065 | red | | | | | | | |
| | OVCYST 066 | | | | | | | | |
| | OVCYST 069 | orange | | | | | | | |
| | OVCYST 067 | | | | | | | | |
| | OVCYST 072 | | orange | | | | | | |
| Malignant (Type I) | OVCYST 031 | | orange | | | | | | |
| | OVCYST 035 | orange | | | | | | | |
| | OVCYST 036 | | red | | | | | | |
| | OVCYST 046 | | | | | | | | |
| | **OVCYST 070 | | | | | red | | | red |
| | OVCYST 071 | | | | | | | | red |
| | OVCYST 074 | | | | | | | | |
| | OVCYST 075 | | orange | | | | | | |
| | OVCYST 076 | | | | | | | | |
| | OVCYST 077 | | red | | | | | | |
| | **OVCYST 078 | | | orange | | | | | orange |
| | **OVCYST 079 | | | | | | yellow | yellow | |
| | OVCYST 080 | | yellow | | | | | | |
| Malignant (Type II) | OVCYST 022 | | | | | | | | red |
| | OVCYST 023 | | | | | | | | red |
| | OVCYST 024 | | | | | | | | red |
| | OVCYST 025 | | | | | | | | red |
| | OVCYST 026 | | | | | | | | red |
| | OVCYST 027 | | | | | | | | red |
| | OVCYST 028 | | | | | | | | red |
| | OVCYST 029 | | | | | | | | red |
| | OVCYST 032 | | | | | | | | red |
| | OVCYST 033 | | | | | | | | red |
| | OVCYST 034 | | | | | | | | red |
| | OVCYST 037 | | | | | | | | red |
| | OVCYST 038 | | | | | | | | red |
| | OVCYST 039 | | | | | | | | red |
| | OVCYST 040 | | | | | | | | red |
| | OVCYST 041 | | | | | | | | red |
| | OVCYST 073 | | | | red | | | | |
| | **OVCYST 081 | | | | | orange | | | orange |

**Figure 3.** Mutated genes found in the cyst fluid samples. Yellow boxes represent mutations with mutant allele frequency (MAF) between 0.1% and 1%; orange boxes represent mutations with MAF between 1 and 10%; red boxes represent mutations with MAF greater than 10% (* indicates patients with insufficient DNA for analysis; ** indicates patients with two detected mutations).

epididymis protein 4) and CA-125 (*Bast et al., 1983*; *Hellström et al., 2003*) (*Table 2*). We defined 'informative' as indicating a need for surgery (i.e., borderline tumors or type I or II cancers). The amount of DNA in cyst fluids was generally, but not significantly, higher in the cysts requiring surgery (p=0.69, *Table 2*), though there were many cysts not requiring surgery that had higher DNA levels than cysts requiring surgery (*Figure 4A*). Similarly, the serum CA-125 levels were significantly higher in cysts requiring surgery (p=0.01, *Table 2*), but there were many cysts not requiring surgery that had higher levels than those requiring surgery (*Figure 4B*). Serum HE4 levels were not correlated with cyst type (p=0.92, *Table 2*; *Figure 4C*). On the other hand, the presence of a mutation was highly informative for the presence of a cyst requiring surgery in the multivariate analysis, as no mutations were found in cysts not requiring surgery (p<0.001, *Table 2*).

## Discussion

Ovarian cancer is the most lethal gynecologic cancer in women. However screening is not recommended by the U.S. Preventive Services Task Force using current diagnostic approaches, which too frequently lead to "important harms, including major surgical interventions in women who do not have cancer" (*Moyer and Force, 2012*). Our study was driven by the underlying principle that clonal cancer driver gene mutations are causative agents of neoplasia and absent in non-neoplastic conditions (*Vogelstein et al., 2013*). We have demonstrated here that driver mutations in ovarian tumors are also present in their associated cyst fluids. Moreover, the mutant allele frequencies in the cyst fluids are relatively high (median 12.6%, IQR of 2.7% to 40.2%), facilitating their detection. There were no mutations detected in the cyst fluids that were not also present in the tumors, and vice versa. Also importantly, no mutation was identified in non-neoplastic cysts or cysts associated with benign tumors. Overall, mutations were detected in a major fraction (87%) of cysts requiring surgery but not in any cyst that did not require surgery (*Table 1*).

Although most (87%) of the 54 cysts requiring surgery had detectable mutations in their fluidic compartment, 7 did not. All of these seven cysts occurred in borderline tumors or type I cancers, while mutations were always (100%) detectable in type II cancers (*Table 1*). There are two potential explanations for our failure to detect mutations in these seven cysts. First, it is possible that the mutant DNA concentration in these cysts was below the level of technical sensitivity of our assay (~0.1% mutant allele fraction). We excluded this possibility by evaluating the tumors themselves: no mutations were detected in any of the tumors from these 7 patients. The second, and therefore more likely explanation, is that our panel of 133 amplicons, containing regions of 17 genes, was not adequate to capture the mutations that were present. Unlike type II cancers, which nearly always contain *TP53* mutations (94% of the type II cancers we studied, for example), the genomic landscapes of type I cancers and borderline tumors are more heterogeneous and not as well studied (*Kurman and Shih, 2010*). Further genetic evaluation of these tumors should facilitate the incorporation of additional amplicons in the panel to reach higher sensitivities. Nevertheless, the 100% sensitivity for type II cancers in our study is highly encouraging, given that these cancers account for over 90% of ovarian cancer deaths.

One limitation of our study is the number of patients evaluated. Though excision of ovarian cysts is one of the most commonly performed surgical procedures, banking of cyst fluids is not common, even in academic centers. Thus, we only had relatively small numbers (n = 22) of non-neoplastic cysts and benign tumors available for study. Even so, the differences in genetic alterations among the various cyst types were striking (*Table 1*). Our study will hopefully stimulate collection and analyses of ovarian cyst fluids that will be able to establish smaller confidence limits around the sensitivities and specificities reported in the current study.

A potential clinical limitation of our approach is the concern by gynecologists that needle puncture of a malignant ovarian cyst leads to seeding of the peritoneum. This concern is based on inconclusive evidence about the dangers of cyst rupture during surgery and is, at best, controversial. A study of 235 patients who had pre-operative ovarian cyst aspirates reported no recurrence in all 7

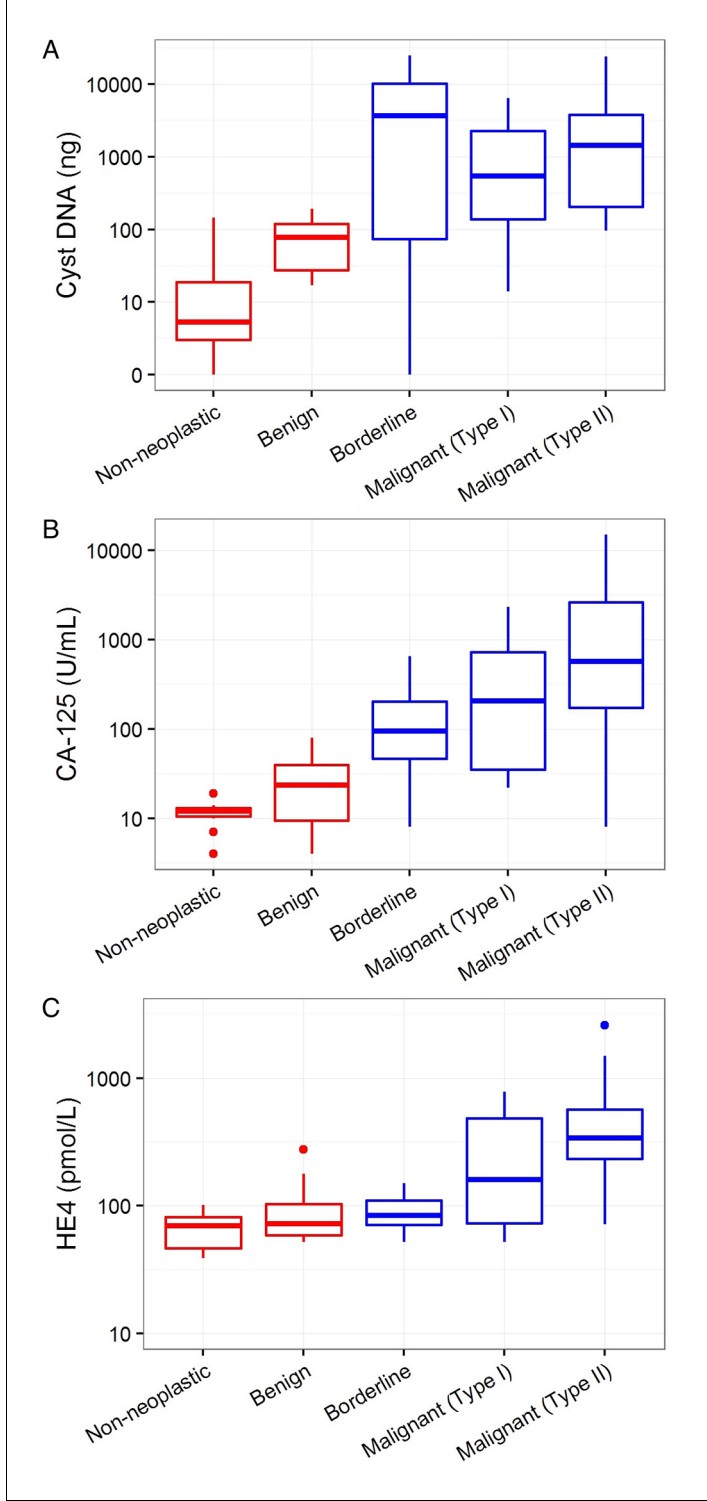

**Figure 4.** Markers associated with the need for surgery. Cyst DNA amount and levels of commonly used ovarian cancer serum biomarkers are plotted according to the cyst type and need for surgery. (**A**) The amounts of DNA in cyst fluids was generally higher in cysts requiring surgery (blue) than those that do not (red), but no significant correlation was found (p=0.69). (**B**) CA-125 levels was significantly higher in cysts that required surgery than those that do not (p=0.01). (**C**) Serum HE4 levels was not correlated with the need for surgery (p=0.92). All values are depicted on a logarithmic scale. P-values were calculated using Firth's penalized likelihood logistic regression in a multivariate model (See Materials and methods).

malignant cases with a mean follow-up of two years (*Mulvany, 1996*). Furthermore, a meta-analysis found no difference in progression-free survival for 2382 early-stage ovarian cancer patients who had experienced intraoperative cyst rupture vs. no rupture (*Kim et al., 2013*). We acknowledge that, under current recommendations, a surgical spill could upstage a localized tumor (i.e. stage 1A to IC) and subject the patient to chemotherapy with its associated morbidity. However, the idea that malignant cysts might shed cancer cells if needle-punctured seems incongruent with the widespread practice of laparoscopic removal of ovarian cysts (*Hilger et al., 2006*). Laparoscopic removal of a cyst carries a significant risk of cyst rupture, conceivably higher than when a tiny needle is inserted under ultrasound-guidance (*Havrilesky et al., 2003*). Finally, malignant pancreatic cysts might be viewed to be as dangerous as malignant ovarian cysts (*Howlader et al., 2014*), yet the standard-of-care for pancreatic cysts involves repeated sampling of cyst fluid through endoscopic ultrasound over many years (*Chang et al., 1997*; *Eloubeidi et al., 2003*). Though pancreatic cysts and ovarian cysts lie in different anatomical compartments, it is encouraging that aspiration of pancreatic cysts is not associated with an increased risk of disease recurrence or mortality in patients with pancreatic cancer (*Beane et al., 2011*; *Ngamruengphong et al., 2013*; *Kudo et al., 2014*; *Ngamruengphong et al., 2015*). Recent advancements in methods of plugging biopsy tracts to prevent tumor cell dissemination, as well as the increased use of intraperitoneal chemotherapy, further lessen the concern of tumor cell dissemination following ovarian cyst aspiration (*Armstrong et al., 2006*; *Tran et al., 2014*; *Tsang et al., 2014*; *Tewari et al., 2015*). On the basis of these observations and recent developments, we believe that ultrasound-guided aspiration of ovarian cyst fluids would likely be a safe and well-tolerated procedure.

As noted in the introduction, as many as twenty patients with benign ovarian cyst lesions undergo surgery for each case of ovarian cancer found (*Buys et al., 2005*). In addition to the psychological impact a potential diagnosis of cancer has on patients, surgery for benign lesions entails considerable cost and morbidity. With the ever increasing sensitivity and use of imaging modalities, the number of patients with incidentally found ovarian cysts is expected to rise. This urgently calls for a diagnostic method that reliably differentiates between the harmless cysts that can be managed expectantly and malignant cysts that require surgical resection. OVA-1 is the only FDA-cleared test to date that aims to distinguish benign versus malignant adenxal mass. It measures levels of five serum markers (CA-125, β-2 microglobulin, apolipoprotein A1, Prealbumin, and transferrin) and is used to stratify patients who should consult a gynecologic oncologist rather than a general gynecologist for surgery. However the test has a specificity of 43% for ovarian cancer, which is even lower than that of CA-125 alone (*Ueland et al., 2011*). While the test might encourage patients with suspected ovarian cancer to seek specialized care, it would not decrease the number of unnecessary surgeries for women with benign adnexal masses.

This study was driven by the need for a biomarker that would help distinguish malignant ovarian tumors from benign lesions and thereby reduce the number of unnecessary surgeries. Such distinction is often difficult based on symptoms and conventional diagnostic criteria. For example, in a large study of 48,053 asymptomatic postmenopausal women who underwent ultrasound examination by skilled sonographers, 8 (17%) of the 47 ovarian cancers that were identified occurred in women with persistently *normal* sonographic findings (*Sharma et al., 2012*). All eight cases were type II cancers, highlighting the potential utility of an additional assay to detect this highly lethal and aggressive type of ovarian cancer. On the other hand, of the 4367 women with *abnormal* sonographic findings, less than 1% of cases proved to have malignancy upon surgery. Furthermore, of the 32 women with borderline or Type I cancers diagnosed, 22 (69%) had a serum CA-125 level within the clinically accepted normal range (≤35 units/mL). In our study, 18 of 18 (100%) type II cancers were detectable by virtue of the mutations found in cyst fluid DNA while none of the 18 benign or non-neoplastic cyst fluid contained such mutations. It is also important to note that the readout of our assay is quantitative and not dependent on the skill level of the reader (in contrast to sonography). Finally, the procedure can be performed minimally invasively in an outpatient setting. The goal of our test is not to replace clinical, radiologic, or sonographic evaluation but to augment them with molecular genetic markers.

Our study, though only proof-of-principle, illustrates one route to improve the management of patients with ovarian cysts. Genetic analysis is not the only such route; proteomics could also provide clues to the correct diagnosis (*Bandiera et al., 2013*; *Kristjansdottir et al., 2013*). One can easily imagine how such additional information could be used to inform clinical practice in conjunction with

current diagnostic approaches. For example, if a cyst contained low amounts of DNA, no detectable mutations, and if the patient had low CA-125 levels, our data suggest that it is very unlikely to be a borderline tumor or malignant lesion. Either no surgery, or laparoscopic rather than open surgery, could be recommended for that patient, even if there were some solid component upon imaging. The option to avoid surgery would be particularly valuable for pre-menopausal women who generally have a low risk of ovarian cancer and might wish to preserve their fertility, as well as patients who are poor surgical candidates. However, our assay in its current format cannot completely rule out malignancy because a fraction of early-stage cancer patients did not have detectable mutations in their cysts. Therefore, patients whose clinical and functional status allows them to undergo surgery and anesthesia might still choose to have a surgical procedure. On the other hand, a minimally invasive test that provides additional, orthogonal information to patients and surgeons could inform their decision about the advisability of surgery.

Our data suggest that a cyst without any solid component upon imaging, and thereby unlikely via conventional criteria to be malignant, should be removed promptly if the cyst fluid contained a *TP53* mutation. Radical, rather than conservative, surgery might be appropriate due to the high likelihood of an aggressive type II cancer. In contrast, if a *BRAF* mutation is identified, the lesion is presumably a borderline or low-grade tumor; thus conservative rather than radical surgery might be sufficient. Furthermore, given that certain types of ovarian cancers (type II) tend to respond well to chemotherapy while others (type I) are relatively chemo-resistant, knowing the type of cancer present prior to surgery based on the mutation profile could help guide decisions regarding the use of neoadjuvant chemotherapy. Validation of the current data in a large, prospective trial will be required before the approach can be seriously considered for clinical implementation in a non-research setting.

## Materials and methods

### Patient samples

Cyst fluids were collected prospectively from women presenting with a suspected ovarian tumor. Patients were diagnosed by transvaginal sonography or computed tomography and admitted for surgical removal of the cyst due to suspicious imaging findings by gynecologic oncology surgeons at Sahlgrenska University Hospital, Gothenburg, Sweden. The study was approved by the ethical board of Gothenburg University and patients provided written consent. According to the approved protocol, 15 to 20 mL of ovarian cyst fluid was collected after removal of the cyst from the abdomen. All samples were immediately put in 4°C for 15–30 min, centrifuged for 10 min at 500 g, and aliquoted into Eppendorf tubes. The fluids were transferred to −80°C, within 30–60 min after collection. All histology was reviewed by board-certified pathologists (*Supplementary file 1*). Ten, 12, 24, and 31 cyst fluid samples from patients with non-neoplastic cysts, benign tumors, borderline tumors, and malignant ovarian cancers, respectively, were assayed in this study.

Plasma HE4 concentrations were determined using a commercial HE4 EIA assay (Fujirebio Diagnostics, PA, USA) and plasma CA-125 levels were measured using the Architect CA 125 II (Abbott Diagnostics, IL, USA). DNA was purified from tumor tissue (either freshly-frozen, or formalin-fixed and paraffin-embedded) after microdissection to remove neoplastic components. DNA was purified from tumors and from 1 mL of each cyst fluid sample using an AllPrep DNA kit (Qiagen, Germany) according to the manufacturer's instructions. Purified DNA from all samples was quantified as previously described (*Rago et al., 2007*).

### Statistical analysis

A Wilcoxon rank-sum test was used to compare the amount of DNA in the cancers and borderline tumors with the amount of DNA in the simple cysts and benign tumors. The fraction of samples detected by tumor-specific mutations in the cyst fluid, as well as their 95% confidence intervals, was calculated for each tumor type (*Table 1*). When the presence of a mutation in the cyst fluid was used to predict the need for surgery, the sensitivity and specificity of the assay, as well as the 95% confidence intervals, were calculated. Firth's penalized likelihood logistic regression was used to quantify the association between molecular features of cyst fluids and the need for surgery (*Table 2*) in a multivariate model. The model predictors included the presence of mutation, log10(ng) of cyst DNA and indicators for normal CA-125 and HE4 values. Normal CA-125 values were defined as <35 U/mL

and normal HE4 values were defined as <92 pmol/L and <121 pmol/L for pre- and post-menopausal women, respectively, according to the cutoffs used at the Sahlgrenska University Hospital. Statistical analyses were performed using the R statistical package (version 3.1.2). Unless noted otherwise, all patient-related values are reported as means ± SD.

## Mutation detection and analysis

DNA from either cyst fluids or tumors was used for multiplex PCR, as previously described (*Kinde et al., 2011*) with the exceptions noted below. One-hundred-and-thirty-three primer pairs were designed to amplify 110 to 142 bp segments containing regions of interest from the following 17 genes: *AKT1, APC, BRAF, CDKN2A, CTNNB1, EGFR, FBXW7, FGFR2, KRAS, MAPK1, NRAS, PIK3CA, PIK3R1, POLE, PPP2R1A, PTEN,* and *TP53*. Primer sequences are listed in *Supplementary file 3*. These primers were used to amplify DNA in 25 µL reactions as previously described except that 15 cycles were used for the inial amplification (*Kinde et al., 2011*). For each sample, three multiplex reactions, each containing non-overlapping amplicons, were performed. Reactions were purified with AMPure XP beads (Beckman Coulter, PA, USA) and eluted in 100 µL of Buffer EB (Qiagen). A fraction (0.25 µL) of purified PCR products were then amplified in a second round of PCR, as described (*Kinde et al., 2011*). The PCR products were purified with AMPure and sequenced on an Illumina MiSeq instrument.

To better distinguish genuine mutations in the samples from artifactual variants arising from sequencing and sample preparation steps, we used Safe-SeqS, an error-reduction technology for detection of low frequency mutations as described (*Kinde et al., 2011*). High quality sequence reads were selected based on quality scores, which were generated by the sequencing instrument to indicate the probability a base was called in error. The minimum quality score requirements were 15 (base call accuracy of 97%) and 20 (base call accuracy of 99%) for each of the fourteen UID bases and mutant base(s), respectively. The template-specific portion of the reads was matched to reference sequences using custom scripts written in SQL and C#. Reads from a common template molecule were then grouped based on the unique identifier sequences (UIDs) that were incorporated as molecular barcodes. Artifactual mutations introduced during the sample preparation or sequencing steps were reduced by requiring a mutation to be present in >90% of reads in each UID family (i.e., to be scored as a 'supermutant'). Only mutations with mutant allele frequency (MAF) above 0.1%, the limit of sensitivity of the assay performed, were considered as positive. Silent or intronic mutations (other than those in the canonical splice sites) were not considered to be mutations because their significance is unknown. In addition, DNA from the peripheral blood lymphocytes of healthy individuals was used as a control to identify potential false positive mutations (see main text). Only supermutants in samples with frequencies far exceeding their frequencies in control DNA samples (i. e., > mean + 5 standard deviations) were scored as positive. The original sequencing data on all mutations listed in *Supplementary file 2* were manually reviewed to confirm that the mutations were correctly called by the Illumina software. Moreover, each of these mutations was validated by an independent PCR and sequencing reaction.

## Acknowledgements

The authors thank J Ptak, L Dobbyn, C Blair, K Judge, and M Popoli for their technical assistance, and E Cook for her artistic contribution.

## Additional information

### Competing interests

IK: is affliated with the PapGene Inc. NJ: is affiliated with the Emmes Corporation. KWK: under agreements between the JHU, Genzyme, Sysmex-Inostics, Qiagen, Invitrogen, and Personal Genome Diagnostics, KWK is entitled to a share of the royalties received by the University on sales of products related to genes and technologies described in this manuscript. KWK is a co-founder of, Personal Genome Diagnostics and PapGene Inc, is a member of the Scientific Advisory Boards of Sysmex-Inostics, Personal Genome Diagnostics, and PapGene, Inc, and owns Personal Genome Diagnostics and PapGene, Inc stock, which is subject to certain restrictions under Johns Hopkins

University policy. The terms of these arrangements are managed by the Johns Hopkins University in accordance with its conflict-of-interest policies. Additionally, KWK is on the Scientific Advisory Board of Morphotek Inc. NP: under agreements between the JHU, Genzyme, Sysmex-Inostics, Qiagen, Invitrogen, and Personal Genome Diagnostics, NP is entitled to a share of the royalties received by the University on sales of products related to genes and technologies described in this manuscript. NP is also a co-founder of, Personal Genome Diagnostics and PapGene Inc., is a member of the Scientific Advisory Boards of Sysmex-Inostics, Personal Genome Diagnostics, and PapGene, Inc., and owns Personal Genome Diagnostics and PapGene, Inc. stock, which is subject to certain restrictions under Johns Hopkins University policy. The terms of these arrangements are managed by the Johns Hopkins University in accordance with its conflict-of-interest policies. LAD: under agreements between the JHU, Genzyme, Sysmex-Inostics, Qiagen, Invitrogen, and Personal Genome Diagnostics, LAD is entitled to a share of the royalties received by the University on sales of products related to genes and technologies described in this manuscript. LAD is a co-founder of, Personal Genome Diagnostics and PapGene Inc., is a member of the Scientific Advisory Boards of Sysmex-Inostics, Personal Genome Diagnostics, and PapGene, Inc., and owns Personal Genome Diagnostics and PapGene, Inc. stock, which is subject to certain restrictions under Johns Hopkins University policy. The terms of these arrangements are managed by the Johns Hopkins University in accordance with its conflict-of-interest policies. BV: under agreements between the JHU, Genzyme, Sysmex-Inostics, Qiagen, Invitrogen, and Personal Genome Diagnostics, BV is entitled to a share of the royalties received by the University on sales of products related to genes and technologies described in this manuscript. BV is a co-founder of, Personal Genome Diagnostics and PapGene Inc, is a member of the Scientific Advisory Boards of Sysmex-Inostics, Personal Genome Diagnostics, and PapGene, Inc, and owns Personal Genome Diagnostics and PapGene, Inc stock, which is subject to certain restrictions under Johns Hopkins University policy. The terms of these arrangements are managed by the Johns Hopkins University in accordance with its conflict-of-interest policies. Additionally, BV is on the Scientific Advisory Board of Morphotek Inc. The other authors declare that no competing interests exist.

## Funding

| Funder | Grant reference number | Author |
| --- | --- | --- |
| Virginia and D.K. Ludwig Fund for Cancer Research | | Bert Vogelstein |
| Banyan Gate Foundation | | Bert Vogelstein |
| Sol Goldman Sequencing Facility at Johns Hopkins | | Bert Vogelstein |
| National Institutes of Health | CA43460 | Bert Vogelstein |
| Cancerfonden | 10-0699 | Karin Sundfeldt |
| BioCARE Program at the University of Gothenburg | 9/14-15 | Karin Sundfeldt |
| Access to Learning Fund | LUA/ALF Grant 70950 | Karin Sundfeldt |
| WeCanCureCancer | | Karin Sundfeldt |

The funders had no role in study design, data collection and interpretation, or the decision to submit the work for publication.

## Author contributions

YW, KS, CM, I-MS, RJK, JS, NS, IK, SS, MF, BK, NJ, KWK, NP, LAD, Conception and design, Acquisition of data, Analysis and interpretation of data, Drafting or revising the article; BV, Conception and design, Acquisition of data, Analysis and interpretation of data, Drafting or revising the article, Contributed unpublished essential data or reagents

## Author ORCIDs

Yuxuan Wang, http://orcid.org/0000-0002-2932-6042
Bert Vogelstein, http://orcid.org/0000-0003-0766-3854

## Additional files

**Supplementary files**
• Supplementary file 1. Patient demographics.
• Supplementary file 2. Mutations identified in cyst fluids and associated tumors.
• Supplementary file 3. Primer sequences used in multiplex assay.

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
