## [Decision Letter]

Thank you for submitting your article "Diagnostic Potential of Tumor DNA from Ovarian Cyst Fluid" for consideration by *eLife*. Your article has been reviewed by three peer reviewers, and the evaluation has been overseen by a Reviewing Editor (Eduardo Franco) and Charles Sawyers as the Senior Editor. Three reviewers were consulted, including Alex Shu-Wing Ng and Michelle Davis who agreed to reveal their identities.

The reviewers have discussed the reviews with one another and the Reviewing Editor has drafted this decision to help you prepare a revised submission.

Summary:

Wang et al. analyzed whether known mutations in ovarian cancer could be found in the patients' ovarian cyst fluid specimens through a highly sensitive sequencing assay. The authors indeed identified specific mutations in borderline tumors, type I+II cancers while no mutations were found in benign or non-neoplastic cystic lesions. The authors conclude that mutant DNA identified in ovarian cyst fluid may guide the diagnostic and therapeutic management of patients with ovarian cysts.

Essential revisions:

Issues related to the study's methodology and interpretation:

1) Methodology needs more details: (i) which tools were used for sequence data alignment? (ii) which tools and what parameters were used for variant calling (including the minimum read count requirement or the minimum variant allele frequency)? (iii) what sequencing depth was achieved per sample? (iv) what was the sequence read quality cut-off used? (v) were variants manually reviewed? (vi) were variant calls limited to missense and nonsense mutations or were no silent mutations observed?

2) Subsection “A multiplex PCR-based assay to identify tumor-specific mutations in cyst fluid samples”: The 5 samples without sufficient DNA should not be included in the analysis. These cases should be excluded instead of included as negative because in clinical practice this would be uninterpretable and thus represents a failure of the technique. While this is small, 5/77 (6.5%) it still should be reported in this manner as low DNA alone, as the authors indicated, is not indicative of a benign cyst and 1 borderline tumor would have been missed with this technique.

3) In Figure 2 what do the points in the non-neoplastic and benign samples represent if no mutations were found?

4) The authors can claim that "the presence of a mutation was highly informative for the presence of a cyst requiring surgery in the multivariate analysis, as no mutations were found in cysts not requiring surgery (P<0.001)" at the end of Results. The authors, however, should also acknowledge that this assay, in the current format, cannot completely rule out that the patient does not require surgery if the assay does not result in any detectable mutations. Other markers like CA125 did not provide much information, as there were many cysts not requiring surgery also had high levels of CA125 than those requiring surgery. The patients might still opt for surgery with a negative mutation finding.

5) Introduction, paragraph two: the authors cite that 5% of patients in the PLCO screening study had malignancy, which is an interesting and important point; however, this screening practice with serial pelvic ultrasounds is not common practice. Does the author have a citation for the risk of malignancy in patients who are symptomatic who present with concerning features on ultrasound as this is a very different population than a screening population?

6) Introduction, paragraph two: What is the citation for the 15% surgical complication rate? With laparoscopic surgery the complication rate for a USO should be far less than 15% and even <15% in laparotomy.

7) The authors address the possibility of using cyst fluid DNA analysis as part of a diagnostic algorithm. The pre-operative US findings for the 77 cases thus would be important to add to this work to better characterize the diagnostic utility. For example, were all of the benign cysts simple appearing and thus by convention a low pre-test probability for malignancy or did they have concerning US features. Including this as well as their pre-op HE4 or CA-125 with the molecular findings would strengthen this work.

Issues related to the clinical implications:

8) The authors do address the major limitation of this study which is the clinical applicability. However, the understanding of when and how this technique could be utilized should be clarified as the current argument is too limited. The authors argue for the safety of US guided needle aspiration of ovarian cysts. While the points are reasonable, the author does not address the problem of upstaging a localized ovarian cancer. While recurrence rates may be the same, a stage IA tumor would not require adjuvant chemotherapy while a IC1 (surgical spill which would include a needle aspiration) would require chemotherapy associated with significant morbidity, impacts on fertility, while not necessarily changing survival or recurrence rate. This point needs to be included in the discussion as a limitation.

9) Furthermore, this technique would require a change in standard of care, and thus would be important to define the exact population that this procedure would benefit. For example, all advanced stage tumors can be excluded as they are clearly malignant by pre-op evaluation. Patients with clearly benign tumors can be excluded- simple appearing cysts on US with normal CA-125. How many patients then are left to whom the pre-op diagnosis is unknown based on abnormal US findings and some elevation in CA-125? I can understand the utility as a confirmatory test for a benign lesion in a patient who by all other parameters appears benign based on the high specificity to rule out malignancy with genomic analysis, but the clinical role here as a screening test really should be flushed out further.

---

## [Author Response]

*Essential revisions:*

*Issues related to the study's methodology and interpretation:*

1) Methodology needs more details: (i) which tools were used for sequence data alignment? (ii) which tools and what parameters were used for variant calling (including the minimum read count requirement or the minimum variant allele frequency)? (iii) what sequencing depth was achieved per sample? (iv) what was the sequence read quality cut-off used? (v) were variants manually reviewed? (vi) were variant calls limited to missense and nonsense mutations or were no silent mutations observed?

We have provided the requested details in the revised manuscript as follows:

i) The sequences were aligned using custom scripts written in SQL and C#.

ii) High-quality reads were grouped into UID families. Only UID families with two or more members (reads) were considered. In order to be scored as a “supermutant”, a mutation had to be present in >90% of reads in each UID family and have a minimum variant allele frequency of 0.1%.

iii) We have added a column in [Supplementary-material SD2-data] to indicate the sequencing depth (in terms of the number of UID families) for each sample or mutation identified. The median number of UID families for samples with sufficient DNA for analysis was 958 (>1916 reads with the requirement of two or more reads per UID family).

iv) The quality score cutoffs were 15 (base call accuracy of 97%) and 20 (base call accuracy of 99%) for the each of the fourteen UID bases and mutant base(s), respectively.

v) The original data on the variants listed in [Supplementary-material SD2-data] were indeed manually reviewed. Moreover, each mutation was confirmed in through an independent PCR and sequencing experiment.

vi) Silent mutations were observed but were not included in the variant calls because we could not be certain that they were functionally significant. Similarly, we excluded intronic mutations other than those at splice junctions for the same reason.

2) Subsection “A multiplex PCR-based assay to identify tumor-specific mutations in cyst fluid samples”: The 5 samples without sufficient DNA should not be included in the analysis. These cases should be excluded instead of included as negative because in clinical practice this would be uninterpretable and thus represents a failure of the technique. While this is small, 5/77 (6.5%) it still should be reported in this manner as low DNA alone, as the authors indicated, is not indicative of a benign cyst and 1 borderline tumor would have been missed with this technique.

We have revised the manuscript and excluded these five cases from further interpretation.

3) In Figure 2 what do the points in the non-neoplastic and benign samples represent if no mutations were found?

We thank the reviewers for pointing out this source of potential confusion. These points represented "zero" mutants, but because of the log scale, we included them on the x-axis, the lowest point possible in log-scale. In the revised Figure 2, we have removed the points in the non-neoplastic and benign samples to more clearly indicate the lack of mutations found in those samples.

4) The authors can claim that "the presence of a mutation was highly informative for the presence of a cyst requiring surgery in the multivariate analysis, as no mutations were found in cysts not requiring surgery (P<0.001)" at the end of Results. The authors, however, should also acknowledge that this assay, in the current format, cannot completely rule out that the patient does not require surgery if the assay does not result in any detectable mutations. Other markers like CA125 did not provide much information, as there were many cysts not requiring surgery also had high levels of CA125 than those requiring surgery. The patients might still opt for surgery with a negative mutation finding.

This is an important point and we have revised the Discussion to include the following statement:

“However, our assay in its current format cannot completely rule out malignancy because a fraction of early-stage cancer patients did not have detectable mutations in their cysts. Therefore, patients whose clinical and functional status allows them to undergo surgery and anesthesia might still choose to have a surgical procedure. On the other hand, a minimally invasive test that provides additional, orthogonal information to patients and surgeons could inform their decision about the advisability of surgery.”

5) Introduction, paragraph two: the authors cite that 5% of patients in the PLCO screening study had malignancy, which is an interesting and important point; however, this screening practice with serial pelvic ultrasounds is not common practice. Does the author have a citation for the risk of malignancy in patients who are symptomatic who present with concerning features on ultrasound as this is a very different population than a screening population?

Thanks for asking us to clarify this point. In the DOvE study of 1455 symptomatic women, 197 women had abnormal features on ultrasound. Of these 197 women, 8 cases (4%) of ovarian cancers were ultimately found (Gilbert L. et al. Lancet Oncol. 2012 Mar;13(3):285-91). We have added this reference to the Introduction.

*6) Introduction, paragraph two: What is the citation for the 15% surgical complication rate? With laparoscopic surgery the complication rate for a USO should be far less than 15% and even <15% in laparotomy.*

In the revised version, we more specifically cite the study on the complication rate (Buys S. et al. Effect of screening on ovarian cancer mortality: the Prostate, Lung, Colorectal and Ovarian (PLCO) Cancer Screening Randomized Controlled Trial. JAMA. 2011 Jun 8;305(22):2295-303). This study reports that “of 3285 women with false-positive results, 1080 underwent surgical follow-up; of whom, 163 women experienced at least one serious complication (15%).”

7) The authors address the possibility of using cyst fluid DNA analysis as part of a diagnostic algorithm. The pre-operative US findings for the 77 cases thus would be important to add to this work to better characterize the diagnostic utility. For example, were all of the benign cysts simple appearing and thus by convention a low pre-test probability for malignancy or did they have concerning US features. Including this as well as their pre-op HE4 or CA-125 with the molecular findings would strengthen this work.

Another important point which we appreciate you raising. All 77 patients had suspicious findings on ultrasound or CT scan in their ovarian cysts, which prompted their surgical removal. We have added this information to the revised manuscript, which as before also includes the pre-op HE4 and CA-125. However, the various imaging criteria used by the surgeons to decide whether to perform surgery were not standardized and changed over the course of the collection period of this study. Some used US (either IOTA or RMI), while others used CT. All we can state is that highly-trained gynecologic oncology surgeons thought each of these 77 patients had worrisome findings mandating surgery.

*Issues related to the clinical implications:*

8) The authors do address the major limitation of this study which is the clinical applicability. However, the understanding of when and how this technique could be utilized should be clarified as the current argument is too limited. The authors argue for the safety of US guided needle aspiration of ovarian cysts. While the points are reasonable, the author does not address the problem of upstaging a localized ovarian cancer. While recurrence rates may be the same, a stage IA tumor would not require adjuvant chemotherapy while a IC1 (surgical spill which would include a needle aspiration) would require chemotherapy associated with significant morbidity, impacts on fertility, while not necessarily changing survival or recurrence rate. This point needs to be included in the discussion as a limitation.

We thank the reviewer for suggesting that we specifically address this point. We also realize that there is some skepticism towards the diagnostic utility and safety of ovarian cyst aspiration in the gynecological community. However we are not the first to encounter such skepticism when presenting a novel diagnostic approach. When colonoscopy was first introduced by Drs. William Wolff and Hiromi Shinya in 1969, it was perceived as both “unnecessary and unduly dangerous” (Wolff W. Am J Gastroenterol. 1989 Sep;84(9):1017-25). This skepticism only began to fade when it was demonstrated that the colonoscope could reveal changes not disclosed by conventional methods at the time. Over the next few decades, and only after many subsequent studies, was the scope’s safety and clinical efficacy demonstrated. Analogously, we have introduced a sort of “molecular scope” for cystic lesions of the ovary that can reveal information unobtainable by conventional methods. As with all such proof-of-principle studies, its clinical safety and efficacy need to be validated in a larger prospective trial. Nevertheless, our data suggest that this new approach, in conjunction with current diagnostic methods, has the potential to improve the management of the vast number of women with ovarian cysts.

To more specifically address the point about Stage I patients, we have modified the revised manuscript to point out that one of the advantages of the approach is that it could lead to planning a less radical surgery in a Stage I patient with *BRAF* or *KRAS* mutation than in a Stage I patient with a *TP53* mutation. The less radical surgery would have a variety of beneficial effects, including maintenance of fertility if desired. On the other hand, we also acknowledge that, under current recommendations, a surgical spill could upstage a localized tumor (i.e. stage 1A to IC) and subject the patient to chemotherapy with its associated morbidity.

9) Furthermore, this technique would require a change in standard of care, and thus would be important to define the exact population that this procedure would benefit. For example, all advanced stage tumors can be excluded as they are clearly malignant by pre-op evaluation. Patients with clearly benign tumors can be excluded- simple appearing cysts on US with normal CA-125. How many patients then are left to whom the pre-op diagnosis is unknown based on abnormal US findings and some elevation in CA-125? I can understand the utility as a confirmatory test for a benign lesion in a patient who by all other parameters appears benign based on the high specificity to rule out malignancy with genomic analysis, but the clinical role here as a screening test really should be flushed out further.

This is an important point that we agree is worthy of further clarification. We do not envision this test as a screening test – only a diagnostic test for women whose diagnosis is doubtful on the basis of conventional clinical and radiologic criteria. The point noted above, i.e., sparing some women radical surgery when the genetic analysis of their cysts indicates that they have a *BRAF* or *KRAS* mutation, is one form of clinical utility. We agree with the reviewer that those with advanced stage tumors would not benefit from this sort of analysis. However, the diagnosis of earlier stage neoplasms is considerably more difficult. We have added the following information to the revised manuscript to make this point more clearly:

“This study was driven by the need for a biomarker that would help distinguish malignant ovarian tumors from benign lesions and thereby reduce the number of unnecessary surgeries. […] Finally, the procedure can be performed minimally invasively in an outpatient setting. The goal of our test is not to replace clinical, radiologic, or sonographic evaluation but to augment them with molecular genetic markers.”